# MITIGATING METASTABLE FAILURES IN DISTRIBUTED SYSTEMS WITH OFFLINE REINFORCEMENT LEARNING

**Yueying Li[1,4], Daochen Zha[2], Tianjun Zhang[3], G. Edward Suh[1], Christina Delimitrou[4], Francis Y. Yan[5]**

[1]Cornell University  [2]Rice University  [3]University of California, Berkeley  [4]MIT  [5]Microsoft Research

## ABSTRACT

This paper introduces a load shedding mechanism that mitigates metastable failures through offline reinforcement learning (RL). Previous studies have heavily focused on heuristics that are reactive and limited in generalization, while online RL algorithms face challenges in accurately simulating system dynamics and acquiring data with sufficient coverage. In contrast, our algorithm leverages offline RL to learn directly from existing log data. Through extensive empirical experiments, we demonstrate that our algorithm outperforms rule-based methods and supervised learning algorithms in a proactive, adaptive, generalizable, and safe manner. Deployed in a Java compute service with diverse execution times and configurations, our algorithm exhibits faster reaction time and attains the Pareto frontier between throughput and tail latency.

## 1 INTRODUCTION

Building reliable cloud services is of paramount importance in distributed systems. With the proliferation of microservices (Gan et al., 2019a), it becomes critical for applications to fortify their services against cascading failures and enduring latency degradation (Gan et al., 2019b; 2021). The microservice architecture is particularly vulnerable to a new form of failure known as *metastable failures* (Huang et al., 2022; Bronson et al., 2021), which exhibit a distinct trait of sustaining effects that hold systems in a degraded state and impede their recovery. Notably, these metastable failures have emerged as the culprits behind significant outages at large internet companies.

We focus on rate limiting to protect systems from such catastrophic failures. Prior work has adopted heuristic-based rate limiting to prevent system overload (Netflix; Kumar; Amazon), but these approaches are reactive, causing the system to endure prolonged capacity degradation induced by metastable failures. Additionally, some strategies suffer from convergence issues in non-stationary environments (Figure 1a)[1]. Lastly, these heuristics require configuring a myriad of system-dependent parameters, thus lacking the ability to generalize across different system contexts.

To address these limitations, we explore learning-based approaches for load control to prevent system metastability. An intuitive solution is to use reinforcement learning (RL), as this problem can be naturally framed as a sequential decision process where the rate limit can be predicted based on the system status at each interval. However, existing online RL algorithms are ill-suited to prevent or mitigate metastable failures in the real world due to the following reasons: *1)* it is difficult to access a high-fidelity simulator that accurately captures the system dynamics; *2)* it is infeasible to explore online and collect *unsafe* data in a real cloud system. To overcome these challenges, we ask: "Is it possible to train a load-shedding policy using only existing log data from cloud services, eliminating the need for extensive tuning of static thresholds for rate limiters?"

Offline reinforcement learning (Levine et al., 2020) emerges as an appealing solution for load control. In this paper, we propose *PolicyShedder*, a load shedding policy designed to proactively mitigate metastable failures using offline RL. PolicyShedder learns directly from native system log data, incurring minimal system overhead. By deploying it on a Java compute service that is prone

---

[1]In this figure, we show a typical case where the heuristic-based load shedder fails to react to system state changes in a timely manner, resulting in cyclic latency spikes and service level objective (SLO) violations.

to metastable failures due to design anti-patterns, we demonstrate that PolicyShedder consistently attains the Pareto frontier between throughput and tail latency, compared with carefully selected heuristics across different contexts (Figure 4), while achieving a 12% lower reaction time, i.e., the time taken to recover from a vulnerable system state with high latency.

## 2 DESIGN OF POLICYSHEDDER

We model the task of load shedding, aimed at mitigating system overload and metastable failures, as a Markov Decision Process (MDP). The formulation is as follows.

**Action space.** The objective of PolicyShedder is to generate a rate limit $\lambda_t$ (number of requests to admit per second) for each pre-configured monitoring time window $\Delta T$. In order to generalize across applications with different service time distributions, we scale the rate limit by Little's law and train an RL model to output $a_t = \lambda_t T_{avg}$, where $T_{avg}$ denotes the average execution time of requests (excluding waiting time in the queue, which can be computed from log data). During deployment, we divide the model's output by $T_{avg}$ to obtain the rate limit $\lambda_t$.

**State space.** The input to PolicyShedder is $s_t = \{\text{Qlen}_t, \text{ewma}(\text{Qlen}_t), \text{Lat}_t, \text{ewma}(\text{Lat}_t)\}$. Here, ewma represents an exponentially weighted moving average over a time interval $[\min\{0, t - n\Delta T\}, t]$, where $n$ is a predefined parameter that captures dependencies in the series of queue lengths (Qlen) and request latencies (Lat) in the log data. An alternative approach involves concatenating a window of historical data into the current observed features. However, there is a trade-off between the state's dimension and the ability to capture dependencies in the time-series data.

**Reward.** We begin with a well-studied metric in the context of congestion control known as Power, defined as the ratio of throughput to delay (Giessler et al., 1978). Gail and Kleinrock proved that the optimal operation point for the network and individual flows is simultaneously attained when Power is maximized (Gail & Kleinrock, 1981). Considering the importance of tail latency in capturing application SLO requirements and metastable signals, we define the reward using aggregated statistics—average throughput and the 95th percentile of tail latency—over a consecutive 10-second period into the future. This approach helps mitigate the impact of temporal load spikes and captures the delayed effect of rewards. Formally, the reward $R_t(\alpha) = \text{Throughput}_{avg}[t : t + n\Delta T] - \alpha \cdot \text{Latency}_{95}[t : t + n\Delta T]$, where $\alpha$ controls the trade-off between throughput and latency. Detailed experimental setups and results are shown in the appendix (A.4–A.9).

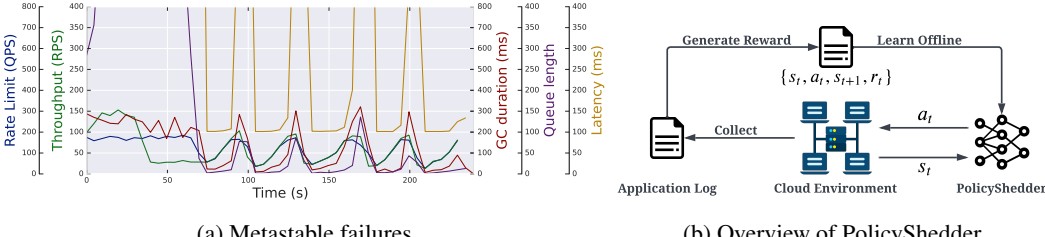

(a) Metastable failures         (b) Overview of PolicyShedder

Figure 1: (a) Illustration of metastable failures. (b) System overview of PolicyShedder. We collect log data from the application and process them using a user-defined function to generate rewards, which are then incorporated into trajectories for training PolicyShedder offline. Once the training completes, PolicyShedder is deployed online to interact with the cloud environment.

## 3 CONCLUSION AND OUTLOOK

We present a practical application of offline reinforcement learning (RL) in cloud systems, with a focus on mitigating critical system failures. This data-driven offline RL-based abstraction provides a valuable tool for constructing intelligent and reliable distributed systems. An extension of our research is to use a multi-agent formulation to enable load shedding for interconnected cloud services with different user requirements, thereby protecting applications from cascading failures.

ACKNOWLEDGMENTS

We thank Zhengyao Jiang for the helpful discussions, especially for providing suggestions on offline RL training methodologies. We also thank Aleksey Charapko for the insightful discussion regarding metastable failures.

URM STATEMENT

The authors acknowledge that at least one key author of this work meets the URM criteria of ICLR 2023 Tiny Papers Track. In this work, Yueying Li meets this criterion.

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

# A APPENDIX

## A.1 RELATED WORK

### A.1.1 OFFLINE RL

In offline RL, a policy is learned from logged data, collected from an environment over a period of time, interaction with the environment is not required. The policy used affects the data distribution collected from an environment. When a policy is learned using an offline dataset, the data distribution when the learned policy is in use differs from the logged data, resulting in a data distribution shift. This remains the fundamental problem with offline RL and several different approaches have been proposed to tackle it.

Offline RL methods can be grouped into two categories in terms of learning and utilizing a model of the environment. Model-based offline RL methods (Janner et al., 2019; Kidambi et al., 2020; Yu et al., 2020; 2021; Jiang et al., 2022; Zha et al., 2022b;a) train a model of the environment using state-action transitions from the logged data. These methods utilize the learned model to generate synthetic episodes, controlled by the policy being trained. The policy parameters are updated using a combination of real episodes (from the logged data) and synthetic ones until convergence. On the other hand, model-free methods (Fujimoto et al., 2019a;b; Kumar et al., 2020; Wang et al., 2020; Kostrikov et al., 2021a) learn a policy that maps states to actions to maximize returns directly. Our methods directly leverage IQL but change the action partition and normalization to be able to generalize to different contexts.

### A.1.2 PERFORMANCE DEBUGGING, ANOMALY DETECTION, AND ROOT CAUSE ANALYSIS

Anomaly detection has been widely studied in machine learning Zha et al. (2020); Lai et al. (2021); Lai et al.; Li et al. (2021a;b; 2020). Recently, anomaly detection has been applied to performance debugging in cloud services (Gan et al., 2021; 2019b; Zhang et al., 2021). Sage uses unsupervised learning and Causal Bayesian Networks for modeling causal relationships among microservices and uses counterfactuals to detect root causes (services and resources) of latency service-level objective (SLO) violation. 93% accuracy in correctly identifying the root cause of QoS violations.

Our problem of mitigating metastable failure is related to but different from performance debugging or mitigating SLO violation. They are related because learning-based methods have great potential to improve the reliability of distributed systems by learning through history logs/traces/metrics. Instead of coding hard-wired mapping of predefined signals/events for an anomaly to mitigation actions (like restarting servers, rebooting, or adding more resources through auto-scaling), the adaptation of anomaly detection and mitigation to new contexts can become an automatic procedure done by machines in a few days with retraining.

They are also different. Performance debugging for SLO violations can be caused by the contention of resources and can be mitigated by resource isolation or auto-scaling. For a metastable failure, it is characterized by a sustaining effect loop either by capacity degradation or workload amplification, which complicates the mitigation strategies.

## A.2 BACKGROUND

We first discussed how we arrive at the right formulation and abstraction. Our problem boils down to setting the right rate or concurrency limit for a service, according to the observed service status (latency, queue size, etc.) When enabled, our limiter will reject excess RPS (request per second) to allow instances to run at a safe and stable state. Our goal is to maximize the throughput and minimize the tail latency of the service and prevent metastable failures from happening when the system enters a vulnerable state[2].

In order to find the right limit of cloud service at the application level, traditionally, people draw wisdom from queuing theory (Harchol-Balter) and manually configured fixed concurrency limits measured via a process of performance testing and profiling. While this provided an accurate value at

---

[2]Extended discussion on how metastable state and vulnerable state are defined can be found in Huang et al. (2022).

that moment in time, the measured limit would quickly become stale as a system's topology changes due to partial outages, auto-scaling, or from code push that impact latency characteristics (url).

**Adaptive rule-based approach:** A natural solution is to use an adaptive rate limiter, or equivalently, a concurrency limiter. An industry example of an adaptive concurrency limiter is from Netflix (Netflix), which draws inspiration from TCP congestion control algorithms (Cardwell et al., 2016; Brakmo et al., 1994; Winstein & Balakrishnan, 2013; Yan et al., 2018), that seek to determine how many packets may be transmitted at a time without incurring timeouts or increased latency. These algorithms, when deployed on the server side, are based on the assumption that latencies are good proxies for queuing. However, when the system is going under a metastable state - each request in the queue could take longer to execute; and moreover, the latency distributions of services could be drastically different. Hence these methods are neither accurate enough to capture system state changes nor generalizable enough to unseen system conditions to prevent or to mitigate metastable failures.

**Adaptive online learning/reinforcement learning:** Similarly, people can use online learning to dynamically adjust the load based on real-time feedback, according to a learned policy (Jay et al.). As an example, using multi-armed bandit or implementing more sophisticated RL-based adaptive online learning to take more states into account has shown promising performance in network congestion control problem (Dong et al., 2018; Mittal et al., 2015). Online learning has shown some promise in other applications too (Google), however, to implement fully online learning algorithms in the real world, it is necessary to collect responses and update configurations in near real-time, which poses significant challenges to the infrastructure. Non-Bayesian online algorithms tend to explore extensively in the initial rounds. This can have a major impact on user experience and lead to SLO violation before the algorithm converges. Furthermore, since the environment is non-stationary, the algorithm may be consistently in the exploration phase, which may cause convergence issues.

**Offline (un-/semi-/supervised) learning:** The third option is to learn a policy from log data without expensive online exploration with a supervised learning approach like behavior cloning (BC) (Bain & Sommut, 1999). Supervised learning is suitable if we can learn a mapping from the state of the system, load shedding action to the utility function of predicted latency and throughput of the service. It is used in congestion control literature. However, this is untenable due to the large state space across different services. Furthermore, inaccurate predictions can cause a feedback cycle known as cascading errors in the long term (Chang et al., 2021). In our problem setup, if we have a slight prediction error that predicts a sub-optimal action, errors can be compounded and lead to more unstable failures.

### A.3 Preliminaries for Offline RL

The RL problem is formulated in the context of a Markov decision process (MDP) $(\mathcal{S}, \mathcal{A}, p_0(s), p(s'|s, a), r(s, a), \gamma)$, where $\mathcal{S}$ is a state space, $\mathcal{A}$ is an action space, $p_0(s)$ is a distribution of initial states, $p(s'|s, a)$ is the environment dynamics, $r(s, a)$ is a reward function, and $\gamma$ is a discount factor. The agent interacts with the MDP according to a policy $\pi(a|s)$. The goal is to obtain a policy that maximizes the cumulative discounted returns:

$$\pi^* = \arg\max_{\pi} \mathbb{E}_{\pi} \left[ \sum_{t=0}^{\infty} \gamma^t r(s_t, a_t) | s_0 \sim p_0(\cdot), a_t \sim \pi(\cdot|s_t), s_{t+1} \sim p(\cdot|s_t, a_t) \right].$$

Off-policy RL methods based on approximate dynamic programming typically utilize a state-action value function ($Q$-function), referred to as $Q(s, a)$, which corresponds to the discounted returns obtained by starting from the state $s$ and action $a$, and then following the policy $\pi$.

**Offline reinforcement learning with implicit Q-learning.** In contrast to online (on-policy or off-policy) RL methods, offline RL uses previously collected data without any additional data collection. Like many recent offline RL methods, our work builds on approximate dynamic programming methods that minimize temporal difference error, according to the following loss:

$$L_{TD}(\theta) = \mathbb{E}_{(s,a,s')\sim\mathcal{D}}[(r(s, a) + \gamma \max_{a'} Q_{\hat{\theta}}(s', a') - Q_{\theta}(s, a))^2], \tag{1}$$

where $\mathcal{D}$ is the dataset, $Q_\theta(s, a)$ is a parameterized Q-function, $Q_{\hat{\theta}}(s, a)$ is a target network (e.g., with soft parameters updates defined via Polyak averaging), and the policy is defined as $\pi(s) = \arg\max_a Q_\theta(s, a)$.

There are three functions to train in IQL:

$$L_V(\psi) = \mathbb{E}_{(s,a) \sim D}[L_2^\tau(Q_\theta(s, a) - V_\psi(s))]$$

where $L_2^\tau(u) = |\tau - \mathbb{1}(u < 0)|u^2$.

The Q-function is trained with the state-value function to avoid querying the actions.

$$L_Q(\theta) = \mathbb{E}_{(s,a,r,a') \sim D}[(r + \gamma V_\psi(s') - Q_\theta(s, a))^2]$$

Finally, the policy function is trained by using advantage-weighted regression.

$$L_\pi(\phi) = \mathbb{E}_{(s,a) \sim D}[\exp(\beta(Q_\theta - V_\psi(s))) \log \pi_\phi(a|s)]$$

## A.4 EXPERIMENTS

Table 1: Rewards of PolicyShedder and the baselines with different average execution times.

| Method | In-distribution | | | Out-of-distribution | | Metastability |
|---|---|---|---|---|---|---|
| | 80 ms | 100 ms | 120 ms | 60 ms | 140 ms | all |
| Best heuristic | $41.17 \pm 0.87$ | $24.99 \pm 0.71$ | $10.47 \pm 0.84$ | $31.00 \pm 3.41$ | $-21.44 \pm 0.92$ | 1/5 |
| BC | $19.91 \pm 12.87$ | $-1379.50 \pm 164.43$ | $-1530.12 \pm 95.98$ | $-1312.36 \pm 429.97$ | $-4685.18 \pm 441.51$ | 2/5 |
| IQL | $50.47 \pm 1.60$ | $\mathbf{30.20 \pm 1.65}$ | $13.59 \pm 1.27$ | $41.06 \pm 1.57$ | $-1.68 \pm 1.41$ | 2/5 |
| TD3+BC | $32.20 \pm 1.65$ | $12.34 \pm 3.24$ | $-43.56 \pm -6.94$ | $-6.12 \pm 6.34$ | $9.42 \pm 2.83$ | 2/5 |
| DT | $24.99 \pm 0.71$ | $23.48 \pm 3.71$ | $13.36 \pm 9.97$ | $-39.12 \pm 89$ | $19.34 \pm 1.70$ | 1/5 |
| CQL | $13.59 \pm 1.27$ | $-4.18 \pm 0.21$ | $-4.50 \pm 2.34$ | $-1.36 \pm 0.97$ | $9.19 \pm 2.78$ | 3/5 |
| PolicyShedder | $\mathbf{53.05 \pm 1.50}$ | $29.14 \pm 2.13$ | $\mathbf{16.59 \pm 0.29}$ | $\mathbf{45.20 \pm 1.22}$ | $\mathbf{32.45 \pm 3.21}$ | 0/5 |

We train PolicyShedder on the log data collected from java application environments with different average execution times and heap sizes, where the execution time of these applications ranges from $\{80, 100, 120\}$ (ms), and the heap size is in $\{192, 256, 512\}$ (MB). The logging policy is a heuristic policy based on TIMELY algorithm (Mittal et al., 2015), which is widely used in datacenter.

The initial version of PolicyShedder is trained with implicit Q-learning (IQL) (Kostrikov et al., 2021b). We used around 500 trajectories with the reward ($\alpha = 0.25$), and each trajectory contains around 4 minutes of log data with monitoring interval as 1 sec. To adapt to sporadic traffic, we choose $\Delta T = \min\{1, \text{time with at least 3 consecutive requests}\}$. However, we found this vanilla offline RL approach is not able to reason about the performance well under distribution shift in transition dynamics (which is the key characteristic of Metastable system, compared to traditional congestion control). Hence, we proposed to use feature normalization and advantage weighted reweighting for our datasets (Hong et al., 2023).

To evaluate the generalizability of PolicyShedder, we test in both in-distribution and out-of-distribution environments. The former uses the same ranges of execution times and the heap sizes, while for the latter, the execution time is selected from $\{60, 140\}$ and only the heap size is selected in the same range. We compare PolicyShedder with several different baselines: **1) Heuristic:** It uses heuristic strategies (Netflix) to control the rate limit in a certain range. [3] We adopt grid-search for the heuristics and report the best result. **2) Behavior cloning (BC) and Offline RL:** BC is an imitation learning algorithm; it uses supervised learning losses to train the policy to imitate the behavioral policies recorded in the log. We include it as it is a common baseline in offline RL research (Kostrikov et al., 2021b; Kumar et al., 2020). We also choose the mostly widely used offline

---

[3]Originally, we use the load-shedder baseline simply as the one in (Netflix). Note that the concurrency limit can be translated to the rate limit because we know the system queue lengths at each time stamp. However, we found that the heuristic is not able to fully prevent the metastable failure from happening due to the delayed nature of load-shedding actions, and requires some prior knowledge of service concurrency limit. Hence, we improve upon the baseline with a stronger version of concurrency control (Mittal et al., 2015).

RL methods, including one-step, pessimistic, and conservative algorithms. Specifically, we implement Conservative Q-learning (CQL) (Kumar et al., 2020), Implicit Q-learning (IQL) (Kostrikov et al., 2021b), TD3+BC (Fujimoto & Gu, 2021), Decision transformer (DT) (Chen et al., 2021))

Table 1 summarizes the results. **Observation 1:** PolicyShedder significantly outperforms the best heuristic, showing the promise of handling metastable failures with offline RL. **Observation 2:** Behavior cloning delivers unsatisfactory performance. This is because the log contains both good and bad behaviors. The supervised policy may have learned undesirable behaviors from the log.

### A.5 HYPERPARAMETERS

We report the hyperparameters used for training the RL agent. Unless otherwise stated, we use grid search for hyperparameter optimization.

Table 2: Hyperparameter of Behavior Cloning (BC).

| Hyperparameter | Value |
| --- | --- |
| Batch size | 100 |
| Regularization factor | 0.5 |

Table 3: Hyperparameter of Implicit Q-Learning (IQL).

| Hyperparameter | Value |
| --- | --- |
| Actor learning rate | $3 \times 10^{-4}$ |
| Critic learning rate | $3 \times 10^{-4}$ |
| Actor optimizer | Adam |
| Critic optimizer | Adam |
| Batch size | 256 |
| N-step TD calculation | 1 |
| Discount factor | 0.99 |
| Target network synchronization coefficiency | 0.005 |
| The number of Q functions for ensemble | 2 |
| The expectile value for value function training | 0.7 |
| Inverse temperature value | 3.0 |
| The maximum advantage weight value to clip | 100.0 |

Table 4: Hyperparameter of TD3+BC.

| Hyperparameter | Value |
| --- | --- |
| Actor learning rate | $3 \times 10^{-4}$ |
| Critic learning rate | $3 \times 10^{-4}$ |
| Batch size | 256 |
| N-step TD calculation | 1 |
| Discount factor | 0.99 |
| Target network synchronization coefficiency | 0.005 |
| The number of Q functions for ensemble | 2 |
| Standard deviation for target noise | 0.2 |
| Clipping range for target noise | 0.5 |
| Alpha | 2.5 |
| Interval to update policy function | 2 |

Table 5: Hyperparameter of Conservative Q-Learning (CQL).

| Hyperparameter | Value |
|---|---|
| Actor learning rate | $3 \times 10^{-4}$ |
| Critic learning rate | $3 \times 10^{-4}$ |
| Learning rate for temperature parameter of SAC | $1 \times 10^{-4}$ |
| Learning rate for alpha | $1 \times 10^{-4}$ |
| Batch size | 256 |
| N-step TD calculation | 1 |
| Discount factor | 0.99 |
| Target network synchronization coefficient | 0.005 |
| The number of Q functions for ensemble | 2 |
| Initial temperature value | 1.0 |
| Initial alpha value | 1.0 |
| Threshold value | 10.0 |
| Constant weight to scale conservative loss | 5.0 |
| The number of sampled actions to compute | 10 |

## A.6 ABLATION STUDIES

Table 6: Model ablation studies.

| | Dropout | Hidden | Dense | BatchNorm | Overhead (s) | In-distribution | OOD |
|---|---|---|---|---|---|---|---|
| MLP - 1 | 0.5 | [32,32] | No | Yes | 0.0032 | $34 \pm 1.65$ | $38.5 \pm 1.34$ |
| MLP - 2 | 0.5 | [16,16] | No | Yes | 0.0031 | $32 \pm 2.09$ | $32.4 \pm 2.23$ |
| MLP - 3 | 0.5 | [8, 8] | No | Yes | 0.0031 | $23.5 \pm 1.94$ | $-19.2 \pm 2.34$ |
| MLP - 4 | 0.2 | [32,32] | No | No | 0.0032 | $26 \pm 1.42$ | $23.4 \pm 1.22$ |
| MLP - 5 | 0.5 | [32,32] | No | No | 0.0031 | $30.5 \pm 0.65$ | $33.3 \pm 3.88$ |
| MLP - 6 | 0.8 | [32,32] | No | No | 0.0031 | $22 \pm 1.23$ | $33.2 \pm 5.32$ |
| MLP - 7 | NA | [32,32] | No | No | 0.0028 | $28.25 \pm 1.21$ | $31.2 \pm 4.20$ |
| MLP - 8 | 0.5 | [32,32,32] | No | Yes | 0.0035 | $35.5 \pm 1.01$ | $45.3 \pm 4.39$ |
| MLP - 9 | 0.5 | [32,32,32,32] | No | Yes | 0.0036 | $34.75 \pm 0.49$ | $43.5 \pm 3.23$ |
| MLP - 10 | 0.5 | [32,32] | Yes | Yes | 0.0033 | $36 \pm 2.65$ | $49.5 \pm 1.32$ |
| Linear | 0.5 | [32] | No | Yes | 0.0028 | $5.75 \pm 1.92$ | $-3.5 \pm 1.21$ |

We now study how each design choice affects performance. In the figure below (Figure **??**), we aggregate the score across the in-distribution and OOD setups.

**Feature choices**: FDC1: choice of multiple feature inputs including memory and CPU utilization; FDC2: without normalization; FDC3: without EWMA features. Surprisingly, we find that additional features like memory and CPU utilization are not helpful in improving the model performance.

**Model choices**: MDC1: use DNN model; MDC2: use LSTM model. We observed that although changing the policy network to a more complex LSTM model seems to be able to get us a slightly higher score, the overhead introduced by the additional complexity overshadows the benefit.

**Look-ahead interval**: We sweep the lookahead window with linear search from $n = 5$ to $n = 15$ but report only three points for the sake of space. The look-ahead window is used for in the reward formulation. In essence, a smaller window makes the system less reactive but may be more sensitive to spurious long requests.

In Table 6, we consider different model architectures, reporting both system overhead and model performance for in distribution and out-of-distribution experiments under different neural architectures. We can see that the overhead is not sensitive to the shape of hidden layers (MLP - 7-9); however, the more the number of layers, the more the agent's performance and sample complexity deteriorates (Sinha et al., 2020). The model performance is sensitive to the dense connections (MLP - 10 vs MLP - 1), especially for OOD environment. Moreover, the wider the hidden layers and the more dropouts, the better the generalizability.

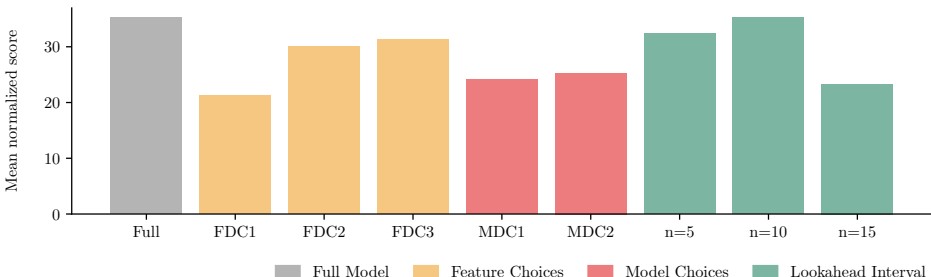

Figure 2: Ablation studies

## A.7 PARSER DETAILS

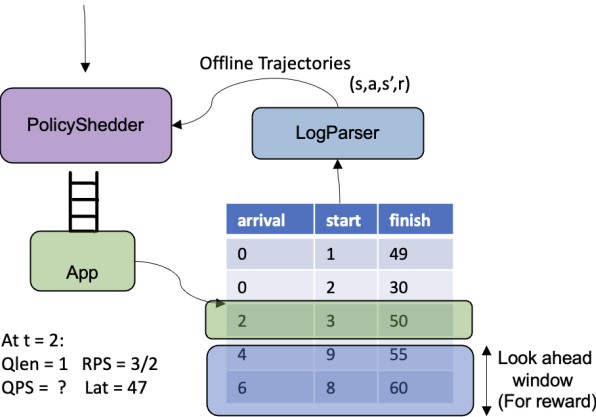

Figure 3: Parser for the log

In Figure 3, we show the details about how we parse the existing request logs to get the trajectories used for offline RL training. The action are indirectly recorded from the log, and the reward can be calculated by taking tail latency and QPS for future look-ahead windows. We further normalize the action and reward according to the service characteristics (i.e. average execution time).

## A.8 OOD PERFORMANCE

In this section, we focus on PolicyShedder's performance under the distribution shift. We make the service have a higher / lower average execution time due to the code upgrade, and report its performance across different system setups. We observe that compared with best heuristic (under linear search) and Behavior cloning which simply learns from the heuristics, it is better at reasoning about the right actions under OOD environment.

Table 7: Out-of-distribution rewards on different execution times and heap sizes for the best heuristic (80, 120, 0.75), behavior cloning, and our PolicyShedder. The best reward is highlighted in boldface, and the second best reward is underlined.

| Method | Execution time 90 | | | Execution time 140 | | |
|---|---|---|---|---|---|---|
| | 192 | 256 | 512 | 192 | 256 | 512 |
| (80, 120, 0.75) | 29.65 | 27.66 | 35.69 | -20.20 | -21.75 | -22.37 |
| Behavior cloning | -708.82 | -1550.01 | -1678.25 | -5086.85 | -4070.34 | -4898.35 |
| PolicyShedder | **43.28** | **39.88** | **40.03** | **-1.67** | **0.05** | **-3.40** |

A.9 VISUALIZATION

Figure 4 shows that PolicyShedder achieves a better tradeoff between throughput and latency compared with the heuristics., i.e., higher throughput and lower latency across all system configurations. The legend tuple shows the average execution time (ms) of the requests in the workload and garbage collection (GC) heap size (MB) configurations.

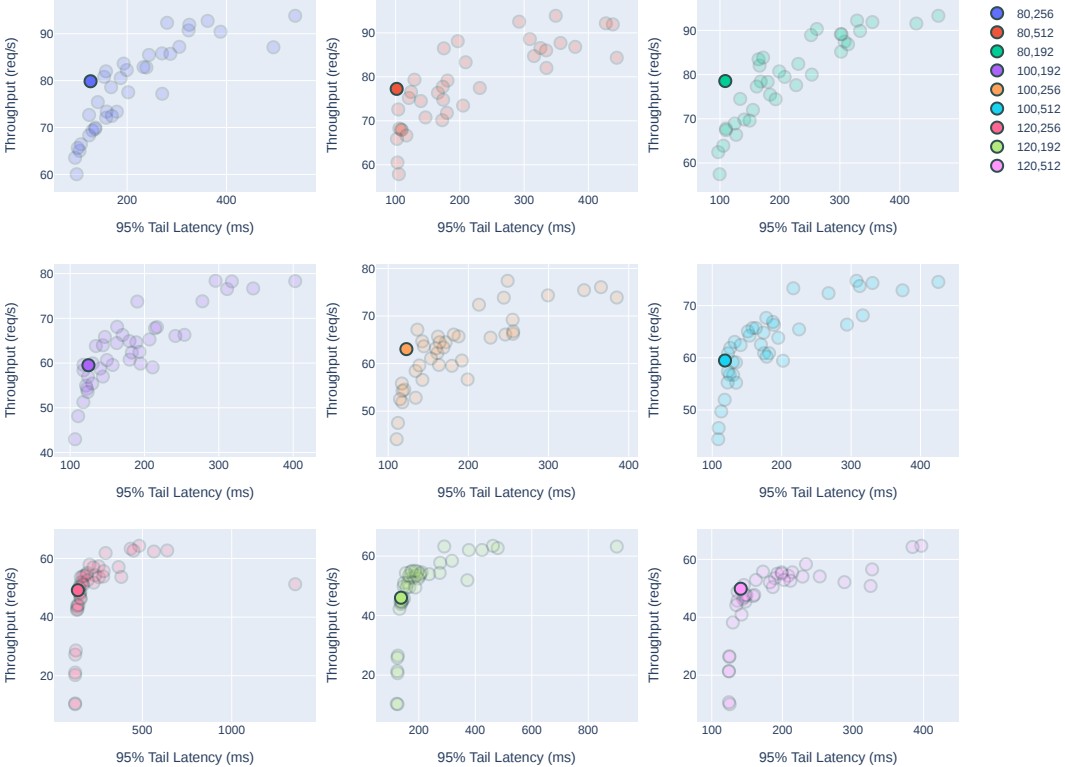

Figure 4: Visualization of PolicyShedder against the heuristics, where PolicyShedder is highlighted as solid selected points.

## A.10 CASE STUDY

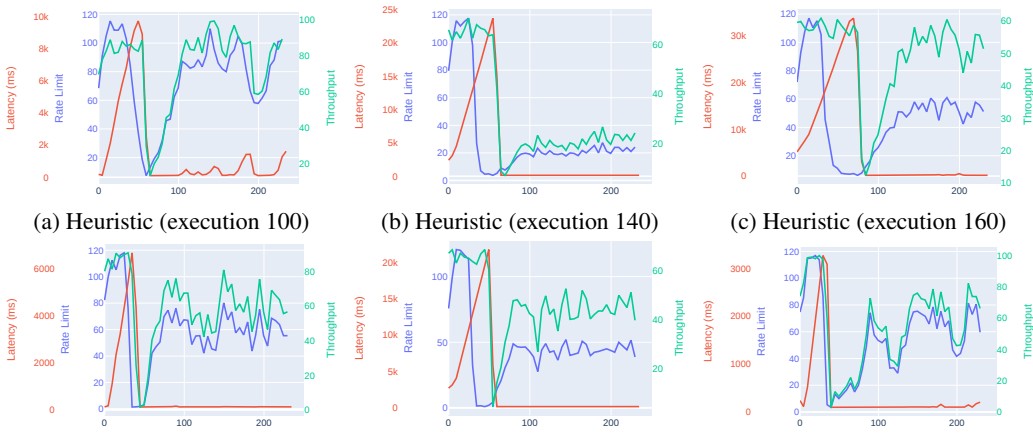

(a) Heuristic (execution 100)   (b) Heuristic (execution 140)   (c) Heuristic (execution 160)

(d) PolicyShedder (execution 100)   (e) PolicyShedder (execution 140)   (f) PolicyShedder (execution 160)

Figure 5: Visualization of the misconfigured heuristic policy when the system has a code upgrade that makes average execution time from 140 ms to 160 ms. The heuristics' reaction time is longer compared with PolicyShedder by average 12%. (a) vs (d) shows how our system is more stable. (b) vs (e) shows a misconfigured heuristic could end up sacrificing the long-term throughput of the service, while our PolicyShedder is less conservative. (c) vs (f) further demonstrate a much faster reaction.

