# OpenReview forum: "Mitigating Metastable Failures in Distributed Systems with Offline Reinforcement Learning"
_ICLR.cc/2023/TinyPapers — Submitted to Tiny Papers @ ICLR 2023_

### Official Review · Reviewer_AW7v · 2023-03-22

**Confidence:** 3

**Summary Of Contributions:**

This paper studies metastable failures in the distributed system. To address the limitation of conventional heuristic-based methods, the authors explore an offline reinforcement learning (RL) based method to predict the rate limit. Extensive experiments demonstrate its superiority and generalization capability.

**Rating:**

High Potential (HP): a submission which meets the reviewing criteria and has potential to make an impact on the field

**Strengths And Weaknesses:**

**Strengths:**

* This paper is indeed well-written. The motivation for adopting offline RL to resolve metastable failures is clear.
* The proposed method seems to be sound. The empirical validation considers in-distribution and OOD settings.

**Weakness:**

* Figure 1 (a) is blurry. Please fix it in the final version.
* The authors mention three key challenges in mitigating metastable failures using RL, without clearly presenting how the proposed method solves each of them.
* Ablation studies are needed. For example, the performance sensitivity on the hyperparameter choices, or the input choices to the RL agent (as described in `State Space`).

**Minor:**

* Is there any transition in your MDP formulation?

* It would be better to illustrate your method using Algorithm (\begin{algorithmic}).

**Overall:**

* *Clarity:* Good. This paper is well-written. It also includes an appropriate discussion of relevant literature in the Appendix.

* *Correctness:* Good. Most of the content in the main paper seems to be correct.

* *Reproducibility:* Fair. It may be hard to reproduce the results without the open-source code.

* *Follows basic requirements:* Yes, the authors follow the ICLR code of conduct.

----

Update: As PC noted, each paper should have an `Abstract` section.

**Suggested Changes:**

Please see the weakness and minor sections.

---

> ### Author Response · Authors · 2023-06-01
> **Thank you for the review**
>
> Dear Reviewer AW7v,
>
> Thank you for your detailed feedback and insightful suggestions.
>
> We appreciate your suggestion to include ablation studies in our work. We agree that evaluating the performance sensitivity on hyperparameter choices and input selections to the RL agent would provide valuable insights. In the final version of the paper, we incorporated these ablation studies to provide a comprehensive analysis of our method's performance are impacted by different design choices, such as the state spaces. Notably, we empirically observed that additional system states are not helpful for the performance, further demonstrating the representation being compact and meaningful towards the set of performance problems we are tackling.
>
> Regarding your minor queries, we understand your suggestion to illustrate our method more clearly, and we will incorporate an algorithm in a full paper version.
>
> Thank you for your time and consideration. We are grateful for your suggestions.

---

### Official Review · Reviewer_DEYV · 2023-04-04

**Confidence:** 3

**Summary Of Contributions:**

The authors in this work propose an offline RL based framework to resolve metastability failures in distributed systems. They demonstrate that their proactive strategy outperforms the traditional heuristic based methods.

**Rating:**

Clear, Correct, and Reproducible (CCR): a submission which meets the reviewing criteria

**Strengths And Weaknesses:**

Strength:
* The authors have clearly presented the problem and proposed a novel method to tackle the problem. The details of the experimental setup are well written and clear for the reader.



Weakness:
* The paper does not seem to have an abstract.

**Suggested Changes:**

* In section 2, rewrite "Power is defined as Power = Throughput/Delay" to "Power is defined as Throuhput divided by Delay."
* Correct spelling of throughput in section 2.
* Please expand abbreviation wherever used eg SLO in section 2, IQL in section 3.
* Add abstract to the paper.

---

> ### Author Response · Authors · 2023-06-01
> **Thank you for the review**
>
> Dear Reviewers,
>
> We would like to express our sincere gratitude for taking the time to review our paper. Your insightful comments and constructive feedback have been immensely valuable in improving the quality of our work. We greatly appreciate your effort in evaluating our submission for ICLR.
>
> We are glad that you found our problem statement clear and our proposed method novel. Your positive feedback on the details of our experimental setup is encouraging, and we are pleased to know that it was well-written and clear for readers.
>
> Regarding the weaknesses you pointed out, we acknowledge your observation that our paper does not currently have an abstract. We apologize for this oversight and fully agree that an abstract is crucial to provide a concise summary of our work. In response to your suggestion, we have incorporated an abstract into the paper. This addition will help readers grasp the key contributions and results of our research at a glance.
>
> Thank you for highlighting the need for some specific revisions in the paper. We have made the following changes as per your suggestions:
>
> In section 2, we will rewrite the sentence "Power is defined as Power = Throughput/Delay" to "Power is defined as Throughput divided by Delay." We also correct the spelling of "throughput" wherever it appears in section 2. Furthermore, we expand the abbreviations used in the paper, such as SLO (Service Level Objective) in section 2 and IQL (Implicit Q-Learning) in section 3. This will ensure clarity and enhance understanding for the readers.
>
> Thank you for your time and consideration. We are grateful for your support in making our research more impactful and accessible.
>
> Best regards,
> Yueying Li

---

### Comment · Area_Chair_YVEW · 2023-06-06
**Check for Archival**

This work meets the threshold for archival, contents the URM statement and is deanonymized.

---

### Meta-Review · Area_Chair_YVEW · 2023-04-02

**Recommendation:** Invite to present
**Confidence:** 3

**Metareview:**

This paper proposes an offline reinforcement learning approach to handle metastable failures in the distributed system, addressing the limitation of conventional heuristic-based methods. Empirical validation on a Java computing service demonstrates its effectiveness and generalization.

This paper only receives one review (AC will adjust the meta-review If another review comes in later). The reviewer acknowledges that the paper meets the CCR standard. However, some issues exist, such as:
* missing abstract section;
* extra ablation studies needed;
* lack of clear explanations about how the proposed method mitigates the limitations of conventional heuristics.

Please also fix the blurry images in the final version.

Overall, based on the review criteria of the ICLR TinyPaper Track, it clearly meets the CCR standard. Please carefully revise and proofread the paper following the reviewer's comments. The AC believes this paper somehow demonstrates a high quality of research. But the AC is not confident about the impact of this paper. The studied topic may be a niche to the ICLR community.

----

Update: we have received another review (CCR). The reviewer suggests some minor revisions, please fix them. My recommendation remains the same.



**Summary:**

This paper proposes an offline reinforcement learning approach to handle metastable failures in the distributed system, addressing the limitation of conventional heuristic-based methods. Empirical validation on a Java computing service demonstrates its effectiveness and generalization.

**Comments And Feedback To The Authors:**

Please carefully revise the paper following the reviewer's comments.

**Reason For Not Giving A Higher Recommendation:**

* Some issues exist. Please refer to the main meta-review.

* The AC believes this paper somehow demonstrates a high quality of research. But the AC is not confident about the impact of this paper. The studied topic may be a niche to the ICLR community.

Therefore, we recommend a poster presentation.

**Reason For Not Giving A Lower Recommendation:**

* This paper is solid and meets the CCR standard. It is a clear acceptance case.

---

> ### Author Response · Authors · 2023-06-01
> **Thank you for the review**
>
> Dear Area Chair,
>
> Thank you for your valuable feedback on our paper. We appreciate your feedback on our high-quality research and will continue to improve this line of work.
>
> Regarding the challenges mentioned in mitigating metastable failures using RL, we apologize for not explicitly addressing how our proposed method tackles each of them. We have explained them more clearly by showing why offline RL is a more favorable approach compared with online RL in the appendix (A7). We will also incorporate some more evaluation in the full version of this work.
>
> We agree that evaluating the performance sensitivity on hyperparameter choices and input selections to the RL agent would provide valuable insights. In the final version of the paper, we have incorporated these in our ablation studies to provide a comprehensive analysis of our method's performance under different configurations.
>
> Thank you again for your valuable feedback, which will significantly improve the quality and clarity of our paper.

---

### Decision · Program_Chairs · 2023-04-07

**Decision:**

Invite to present

**Comment:**

Please add your URM statement.